# Dynamic neurogenomic responses to social interactions and dominance outcomes in female paper wasps

**Floria M. K. Uy** [ID][¤]**, Christopher M. Jernigan** [ID]**, Natalie C. Zaba, Eshan Mehrotra** [ID]**, Sara E. Miller** [ID]**, Michael J. Sheehan** [ID]*****

Laboratory for Animal Social Evolution and Recognition, Department of Neurobiology and Behavior, Cornell University, Ithaca, New York, United States of America

¤ Current address: Department of Biology, University of Rochester, Rochester, New York, United States of America
* msheehan@cornell.edu

**Data Availability Statement:** Raw sequence data are available in the NCBI Short Read Archive under PRJNA705303. All numerical data used in analyses is available as supplemental files.

## Abstract

Social interactions have large effects on individual physiology and fitness. In the immediate sense, social stimuli are often highly salient and engaging. Over longer time scales, competitive interactions often lead to distinct social ranks and differences in physiology and behavior. Understanding how initial responses lead to longer-term effects of social interactions requires examining the changes in responses over time. Here we examined the effects of social interactions on transcriptomic signatures at two times, at the end of a 45-minute interaction and 4 hours later, in female *Polistes fuscatus* paper wasp foundresses. Female *P. fuscatus* have variable facial patterns that are used for visual individual recognition, so we separately examined the transcriptional dynamics in the optic lobe and the non-visual brain. Results demonstrate much stronger transcriptional responses to social interactions in the non-visual brain compared to the optic lobe. Differentially regulated genes in response to social interactions are enriched for memory-related transcripts. Comparisons between winners and losers of the encounters revealed similar overall transcriptional profiles at the end of an interaction, which significantly diverged over the course of 4 hours, with losers showing changes in expression levels of genes associated with aggression and reproduction in paper wasps. On nests, subordinate foundresses are less aggressive, do more foraging and lay fewer eggs compared to dominant foundresses and we find losers shift expression of many genes in the non-visual brain, including vitellogenin, related to aggression, worker behavior, and reproduction within hours of losing an encounter. These results highlight the early neurogenomic changes that likely contribute to behavioral and physiological effects of social status changes in a social insect.

## Author summary

Aggressive interactions often create inequalities–some individuals win while others lose. Winning versus losing can lead to large physiological differences between individuals,

**Funding:** MJS received DP2-GM128202 from the National Institutes of Health (https://www.nih.gov/). The funders had no role in study design, data collection and analysis, decision to publish, or preparation of the manuscript.

**Competing interests:** The authors have declared that no competing interests exist.

including different neurogenomic profiles between winners and losers. How this information about contest outcome leads to distinct neurogenomic profiles is poorly understood. Here we examine gene expression in response to aggressive social encounters in paper wasps, which naturally form dominance hierarchies on their nests in the wild. Shortly following encounters winners and losers have similar expression profiles, likely because similar mechanisms are engaged by social experiences. Four hours later, we find divergent neurogenomic profiles between winners and losers, with losers showing larger shifts in expression compared to winners. Many of the most dynamically expressed genes have been previously associated with dominance and caste differences in paper wasps showing how a single interaction can engage many of the same genomic networks that are involved in mediating more dramatic differences in queen-worker behavioral differences are also involved in responses shortly following social interactions.

## Introduction

Social interactions can give rise to a range of immediate as well as long-lasting effects on behavior and physiology [1–4]. Regardless of the nature of the interaction or the outcome, social experiences are expected to have a number of shared effects on the physiology of those involved. Processing social information may depend on multiple cues or signals, which are likely to be processed by similar brain regions and genes within [5–7] and across species [8, 9]. In addition to social information processing, interactions can increase rates of activity and movement, especially in relation to courting or fighting [2, 8]. In recent years there has been a growing number of gene expression studies examining the neurogenomic responses to social interactions across a range of taxa including honeybees, mice and sticklebacks [8, 10, 11], finding shared elements of neurogenomic responses immediately following social challenges. Indeed, at the level of neural firing patterns, social interactions give rise to similar patterns of neural activity in bats and mice [6, 7]. Similar initial patterns of neural activation and transcriptomic changes, however, give way to divergent effects depending on the outcome of encounters. Longer-term consequences of social interactions depend on the nature and outcome of the encounters. Winning versus losing typically cause different physiological and behavioral responses [12–18]. Over repeated interactions, this can lead to profound differences in behavior, physiology, life expectancy, and fitness [4, 19–22]. Divergent social outcomes lead to different physiological responses, which may be initiated by differences in neurogenomic responses shortly following an interaction.

There have also been studies examining the effects of winning and losing rather than simply the response to social challenge *per se*. In zebrafish, socially driven transcriptional changes require individuals to assess the outcome of the interaction [23] (i.e., did they win or lose). In sub-social carpenter bees, repeatedly winning or losing staged contests gives rise to distinct neurogenomic profiles [16, 24]. In the ant *Harpegnathos saltator*, workers compete for reproductive openings upon the removal of the queen and within a few days individuals have divergent neurogenomic profiles depending on their trajectory toward either staying as a worker or becoming a reproductive gamergate [25]. Similar divergence in social behavior and neurogenomic profiles are seen among *Polistes dominula* paper wasp workers fighting for the dominant breeding position upon queen removal [26, 27]. Collectively, these studies demonstrate that social interactions can have immediate effects on neurogenomic profiles and that repeated interactions can have longer-term consequences for patterns of transcription in the brain that differ for winners and losers or higher- versus lower-ranking individuals. Understanding how

transcriptional patterns change over time in response to different social interactions and across different taxa will help us to more clearly link social outcomes to short and long-term physiological changes.

Understanding the dynamic changes that occur between initial responses and subsequent divergence between winners and losers will help link these two areas of research. Studies examining the temporal dynamics of transcriptional responses to social challenge in stickleback and mice over the course of a few hours highlight the transient and dynamic nature of transcriptional responses [10, 11]. Detailed work on the early transcriptional responses to fighting between pairs of male beta fish demonstrates that fighting individuals have shared transcriptomic responses within the first hour after fighting [5]. The studies mentioned above have looked at dynamic responses to a social challenge from territorial or nest intrusions or more established winner-loser effects. The dynamics by which interacting individuals develop divergent transcriptomic responses over the course of a few hours has received less attention.

Here we examine the dynamic neurogenomic responses to social interactions in female *Polistes fuscatus* paper wasp foundresses over the course of four hours following a staged social interaction. Paper wasps are primitively eusocial insects in which females found new nests each spring after overwintering [28]. Social interactions among paper wasp foundresses lead to profound physiological differences between dominants and subordinates. Nests are initiated by a single foundress or small groups of foundresses, who form an aggression-based dominance hierarchy, which determines the extent of work and egg-laying [29, 30]. Polistine foundresses have aggressive interactions in both the pre-nesting stage as well as on the nests, where they interact aggressively with co-foundresses as well as occasional usurpers [31–34]. Wasps also reliably show aggression to other individuals in neutral arenas, providing a convenient method for studying the effects of aggression in a controlled setting [35–37]. Previous work has shown that *Polistes* foundresses respond rapidly to aggressive encounters by modulating juvenile hormone [18], though genome-wide transcriptomic responses have yet to be examined immediately following aggressive interactions. In established co-foundress associations, dominant and subordinate foundresses show differential expression of genes associated with aggressive behavior [38]. By comparing the temporal shifts in gene expression between winners and losers, we can potentially identify genes that are associated with the early stages of dominance hierarchy formation in paper wasps. Additionally, such analyses may generate more general insights into the neurogenomic processes by which social interactions lead to divergence in behavior and physiology.

The neurogenomic responses to social interactions in *P. fuscatus* are also of interest because this species recognizes individuals based on variable facial features [39, 40]. Individual recognition appears to mediate dominance interactions among groups in the lab and on natural nests [36, 39]. Individual recognition is not present in other closely related species of paper wasps [40, 41], suggesting the trait has evolved relatively recently [42]. Neurogenomic responses to operant conditioning related to face-learning have been previously studied [43], but their neurogenomic responses to social interactions have not been investigated. Wasps are known to form long-term memories of those they have interacted with [44], so examination of neural transcriptomes a few hours after the interaction has the potential to reveal insights into the neurogenomic responses related to social memory, as long-term memory formation occurs hours after initial learning has occurred [45]. Visual facial recognition is a notable feature of *P. fuscatus* from Ithaca, NY [39], so we were also interested in the relationship between visual processing and social interactions. Previous studies of eye morphology have suggested that the visual system may have evolved to facilitate individual recognition [46] and visual brain regions are developmentally sensitive to social isolation [47]. Additionally, analyses of selection in *P. fuscatus* identified visual processing genes as enriched as recent targets of positive

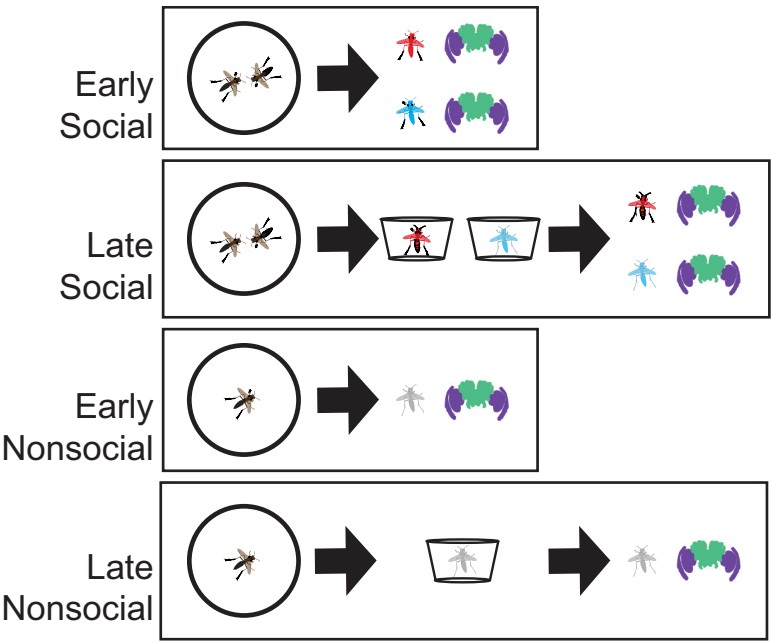

**Fig 1. Overview of experimental design and RNAseq data.** (A) The experiment consisted of generating two groups of wild-caught wasps that either engaged in a recent social experience or remained nonsocial. Half of each group was sacrificed at the end of a 45-minute interaction period with the other half held in individual containers for 4 hours until they were then sacrificed. RNA was extracted separately from the combined optic lobes (purple) and the remainder of the brain, called 'brain' throughout (green). In other figures, we show the part the tissue the data is derived from with the relevant icon. Here and in subsequent figures, red wasp symbols are used to indicate winners, blue wasp symbols for losers, and grey wasps for control individuals that did not have social interactions.

selection in this species [42], likely in relation to social interactions and individual recognition. Given the importance of vision in social interactions for this species, we examined the effects of social interaction on the optic lobes, which contain early visual processing neuropils, as well as the non-visual brain (Fig 1, hereafter 'optic lobe' and 'brain').

We designed an experiment to examine the dynamic neurogenomic responses shortly after social interactions in the optic lobe and brain (Fig 1). Wasps were filmed in a neutral arena while paired with another weight-matched individual or alone (S1 Movie). To better understand the temporal dynamics of neurogenomic responses in the hours following a social interaction, we looked at transcriptomes at two time points: immediately following a 45-minute interaction and after 4 hours of separation back in the wasps' original housing containers (Fig 1). For each of the two time points we generated 15 social trials (n = 30 wasps) and 15 nonsocial trials (n = 15 wasps). The social trials tend to be aggressive, generating winners and losers for each trial (n = 15 winner and n = 15 losers at each time point). In total we examined the behavior of 60 wasps in social trials and 30 wasps went through nonsocial trials, though the numbers for RNAseq analyses were smaller due to some samples not generating libraries or sufficient sequencing (Table 1). In the grander scheme of paper wasp dominance relationships, both of these timepoints are very early in the time course over which a dominance hierarchy would be formed. For ease of distinguishing between the samples we refer to those taken immediately at the end of a 45-minute interaction as 'early' and those at 4 hours as 'late'.

Using the RNAseq data from paper wasp foundresses, we address multiple questions. (1) How does the magnitude of neurogenomic responses differ between earlier versus more central brain regions? To the extent that responses are driven by the processing of social outcomes

**Table 1. Sample Numbers Used in Analyses.**

| Sample Numbers Used in Analyses | | | |
|---|---|---|---|
| | Behavioral Data | RNAseq: Non-Visual Brain | RNAseq: Optic Lobe |
| **Early Social** | 30 | 24 | 27 |
| Winners | 15 | 7 | 9 |
| Losers | 15 | 10 | 10 |
| **Late Social** | 30 | 20 | 24 |
| Winners | 15 | 8 | 8 |
| Losers | 15 | 5 | 7 |
| **Early Non-Social** | 15 | 12 | 13 |
| **Late Non-Social** | 15 | 10 | 9 |

NB: Winner and Loser numbers only include individuals from trials with 10+ aggressive acts

rather than simply response to social stimuli, we may expect larger and or more dynamic changes in more central compared to earlier brain regions, such as the optic lobe. (2) Given that paper wasps learn and remember the identities of wasps they interact with [44], is there a detectable neurogenomic signature related to memory in paper wasps following interactions? (3) How does social outcome influence the dynamics of neurogenomic responses over the course of a few hours? Recent studies suggest similar neural responses among individual during or right after social interactions [5–7], whereas others demonstrate divergent outcomes over the course of days [16, 24, 25, 27]. Therefore, we may predict that initial neurogenomic responses will be more similar immediately following social interactions and that winners and losers will diverge transcriptionally over time.

## Results

### Measurement of behavioral interactions

We compared the amount or intensity of aggression between wasps across trials. On average there were 33.13 ± 12.58 aggressive acts per trial. The overall intensity of aggression in neutral arena trials tends to be relatively mild compared to the fights that can happen when defending a nest from a usurper [28, 31], though the amount of aggression observed here is in line with other published studies of aggression between *P. fuscatus* foundresses in a neutral arena [44, 48]. Across trials, the aggregate rates of aggression were highest in the beginning and declined over time (Fig 2A). We defined aggression both in terms of the number of aggressive acts as well as an aggression score taking into account differences in the relative severity or intensity of aggressive acts, following previous research on aggression in staged encounters between wasps [41, 48]. As would be expected, the two measures are highly correlated (Fig 2B and S1 Table, linear model, $F_{1,28}$ = 235.9, $P$ = 3.6e-15, $r^2$ = 0.89). In all cases, one individual was more aggressive by one or more measures, which we termed the 'winner'. There are no cases where losers had higher values of directed aggression than winners in terms of the number of acts, summed aggression score, or aggression index. We found a positive relationship between the aggressiveness of winners and losers in terms of the number of aggressive acts (Fig 2C, linear model, $F_{1,28}$ = 90.2, $P$ = 2.97e-10, $r^2$ = 0.76), summed aggression scores (linear model, $F_{1,28}$ = 54.6, $P$ = 4.75e-8, $r^2$ = 0.66), and the aggression index (linear model, $F_{1,28}$ = 4.39, $P$ = 0.045, $r^2$ = 0.13). The correlation in the amount of aggression between the two wasps within a trial suggests that higher levels of aggression may come about through a process of escalation. The variation in the total amount and intensity of aggression across trials means that though 'winners'

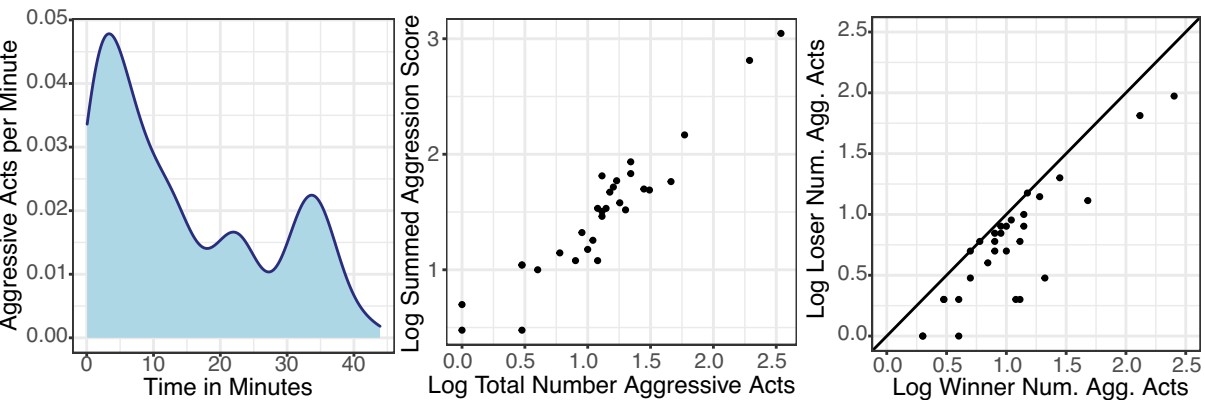

**Fig 2. Behavioral interactions.** (A) Trials were characterized by initially increasing and then decreasing rates of aggression. Most of the aggression within the trials took place in the first few minutes. (B) The number of aggressive acts and the overall intensity as measured by the summed aggression are highly correlated. (C) Across trials, one individual was more aggressive in terms the number of acts, the summed intensity or acts, or an index of the average intensity of the acts. Here the number of acts between the more aggressive individuals (i.e., winners) are shown on the x-axis relative the number of aggressive acts committed by the less aggressive individuals (i.e. losers) on the y-axis. The line shows the 1:1 relationship. The positive relationship between aggression between individuals in the trials suggests are possible role for mutual escalation in shaping contest behavior.

were more aggressive than 'losers' in each trial, binary outcomes are not strictly indicative of relative differences in individual experiences across trials. In other words, some winners received more aggression than some losers and some losers were more aggressive than some winners.

## Social interactions generate stronger and more dynamic neurogenomic responses in the brain compared to optic lobes

We first compared RNAseq data from 139 samples (Table 1) in DESeq2 with a model that included tissue (optic lobe or brain), whether or not wasps had been placed in a social or control trial, and time of sacrifice as separate categorical main factors. Optic lobes and the brain show distinct transcriptional profiles that are well-separated in a principal component analysis (PCA) (Fig 3A). Furthermore, social and non-social samples are better separated in a PCA of the entire dataset among brain samples compared to the optic lobes (Fig 3A). Though the social and non-social groups do not form two distinct clusters as has been found in other transcriptomic studies related to social behavior in other species (e.g. ref 5). Based on the output of the PCA, we examined the effects of social experience, time of sacrifice, and their interaction across the first 15 PCs generated by PCA of the entire datasets (69.98% of cumulative variance explained) separately for the brain and optic lobe (S2 Table). In the brain samples, social experience is a significant predictor of PC1 (27.75% of variance explained, $F_{1,62} = 7.76$, $P = 0.0071$) and PC5 (3.5% of variance explained, $F_{1,62} = 10.56$, $P = 0.0019$). In the optic lobe samples, social experience explained variation in PC10 (1.34% of variance explained, $F_{1,69} = 6.64$, $P = 0.012$) and PC15 (0.92% of variance explained, $F_{1,69} = 4.24$, $P = 0.04$). We find a similar pattern when conducting separate PCAs for each tissue type (S2 Table). Thus, social experience affects higher level principal components in the brain compared to the optic lobes, suggesting an overall larger impact of social behavior on the transcription of more central and higher order brain integration centers compared to earlier visual processing centers in the optic lobe.

We examined a model in DESeq2 with tissue, time of sacrifice, and social experience as factors. We identified 4937 differentially expressed genes (DEGs) between the brain and optic

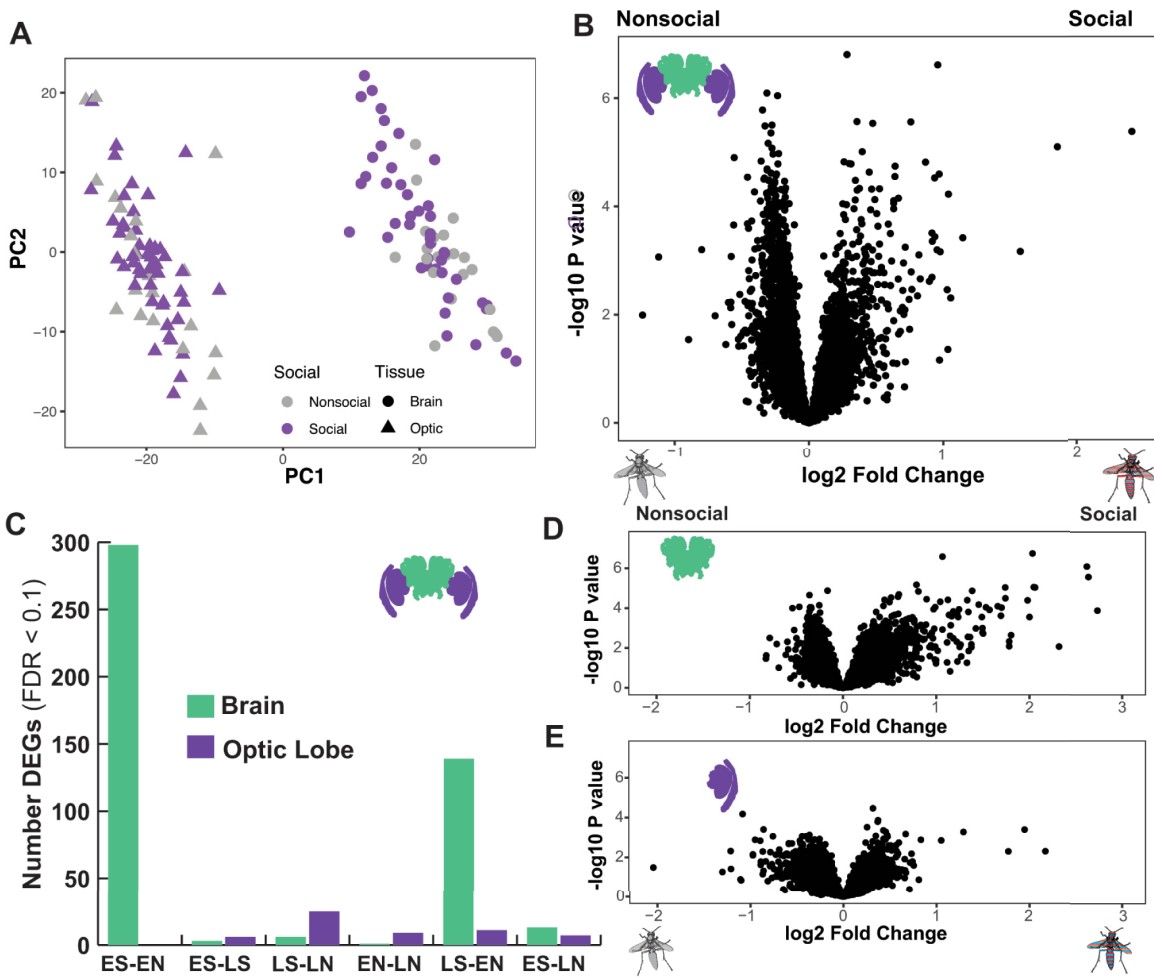

**Fig 3. Social interactions influence neurogenomic signatures more in the central brain than in the optic lobes.** (A) Brain region is the strongest separator of the data in a principal component analysis. Recent social experience has a stronger effect on the samples from the central brain compared to the optic lobes in the PCA. (B) Hundreds of genes (n = 742, FDR < 0.1) are differentially expressed as a result of social interactions in the combined dataset of both time points and tissues. (C) The effects of social interactions are stronger in the central brain compared the optic lobes. At both early and late time points there are hundreds of genes differentially expressed (FDR < 0.1) between social and nonsocial groups. The following codes are used in the axis legend: ES = early social, EN = early nonsocial, LS = late social, LN = late nonsocial. (D-E) The volcano plots show the log2 fold change between social (up) and nonsocial (down) on the x-axis and the -log10 P value on the y-axis. The red and blue striped wasp symbol indicates that the data includes all socially interacting wasps. Panel D shows data for the central brain, while panel E shows data for the optic lobes.

lobes consistent with different cellular compositions between the two tissues. Time of sacrifice showed a minor effect on overall patterns of gene expression with 73 DEGs. In contrast, social experience had a more pronounced effect on patterns of gene expression, with 742 DEGs (Fig 3B). We next considered a model comparing each group based on brain region, time and social experience as a single combined factor (e.g., brain_early_social v. brain_early_nonsocial). Consistent with PCA (Fig 3A), the comparisons reveal a stronger effect of social interactions on the brain compared to the optic lobe (Fig 3C). The results are qualitatively similar when examining the effects of social experience and time on brain and optic lobe datasets separately (Fig 3D and 3E and S3 Table).

## Socially responsive genes are enriched for memory-related functions

We identified 61 overrepresented GO terms (P< 0.01) among the 742 social DEGs in the full model with brain region, social experience, and sampling time as separate categorical factors. Many of the GO terms deal with membrane transport, calcium signaling, synaptic transmission or behaviors, which are to be expected given that we analyzed a neurogenomic dataset related to adult behavior (S4 Table). A few of the enriched categories, however, suggest neurogenomic processes supporting social behavior in *Polistes* wasps. For example, genes annotated as being involved in cholinergic synaptic transmission are overrepresented among socially responsive genes (GO:0007271, P = 0.0015), indicating that cholinergic neurons may play a role in the aggressive encounters between the wasps. Recent work in *Drosophila* has implicated cholinergic signaling in aggression in both males and females [49–51], highlighting potentially shared mechanisms related to aggressive interactions across taxa.

Female *P. fuscatus* learn and remember the identity of other wasps from previous interactions [44] or even outcomes of fights among other individuals they have seen interacting [48]. Behavioral experiments have demonstrated both short and long-term memories of individuals [44, 48], suggesting that signatures of both processes may be enriched among differentially regulated genes. Therefore, we were specifically interested in testing the *a priori* hypothesis that transcriptomic signatures following social interactions would be enriched for differential expression of genes related to memory. Indeed, genes annotated with functions in anesthesia-resistant memory (GO:0007615, P = 3.6e-5) and long-term memory (GO:0007616, P = 0.009) are enriched among socially responsive genes. A puzzling feature of the expression of genes annotated with memory functions is that they frequently appear to be down regulated among individuals in the social compared to nonsocial treatments (S1 Fig). Memory formation is a dynamic process with multiple steps in which genes are up- and down-regulated at different times [52] and the observed down-regulation may reflect aspects of that dynamic process. Likely relevant to memory formation, socially responsive genes are enriched for functions relating to mushroom body development (GO:0016319, P = 0.00055), synaptic target recognition (GO:008039, P = 0.00029), and regulation of synaptic plasticity (GO:0048167, P = 0.0051).

## Similarities and differences in winner and loser neurogenomic responses

Individual wasps had different experiences of social interactions depending on whether or not they were the individual giving or receiving more aggression–i.e., whether they were the winner or the loser of the encounter. Therefore, we considered the neurogenomic responses separately for the individuals that won or lost the social encounters compared to those that had not been involved in a social interaction. For these subsequent analyses, we removed the least aggressive trials from the dataset (< 10 total aggressive acts), as the interpretation of aggressive outcomes in the context of trials with particularly low aggression is less clear biologically. Excluding the least aggressive trials results in more differentially expressed genes detected in the brain in response to social encounters (all data = 348 DEGs, aggressive trials only = 433 DEGs), despite the reduced sample size. This suggests that removing the least aggressive trials from the dataset provides a cleaner signal of the effects of aggressive outcomes.

In a model considering encounter outcome, tissue (i.e. brain or optic lobe), and time as main factors, we found overall similar numbers of DEGs for tissue (4435 DEGs) and time (22 DEGs) as with the model based on social experience. Both winners and losers had hundreds of differentially expressed genes compared to nonsocial individuals, though the neurogenomic response appears to be stronger in losers (Fig 4A, winners = 217 DEGs, losers = 484 DEGs). When directly compared to each other, winners and losers show no significant differences in gene expression based on the FDR < 0.1 threshold in DESeq2. Even considering less restrictive

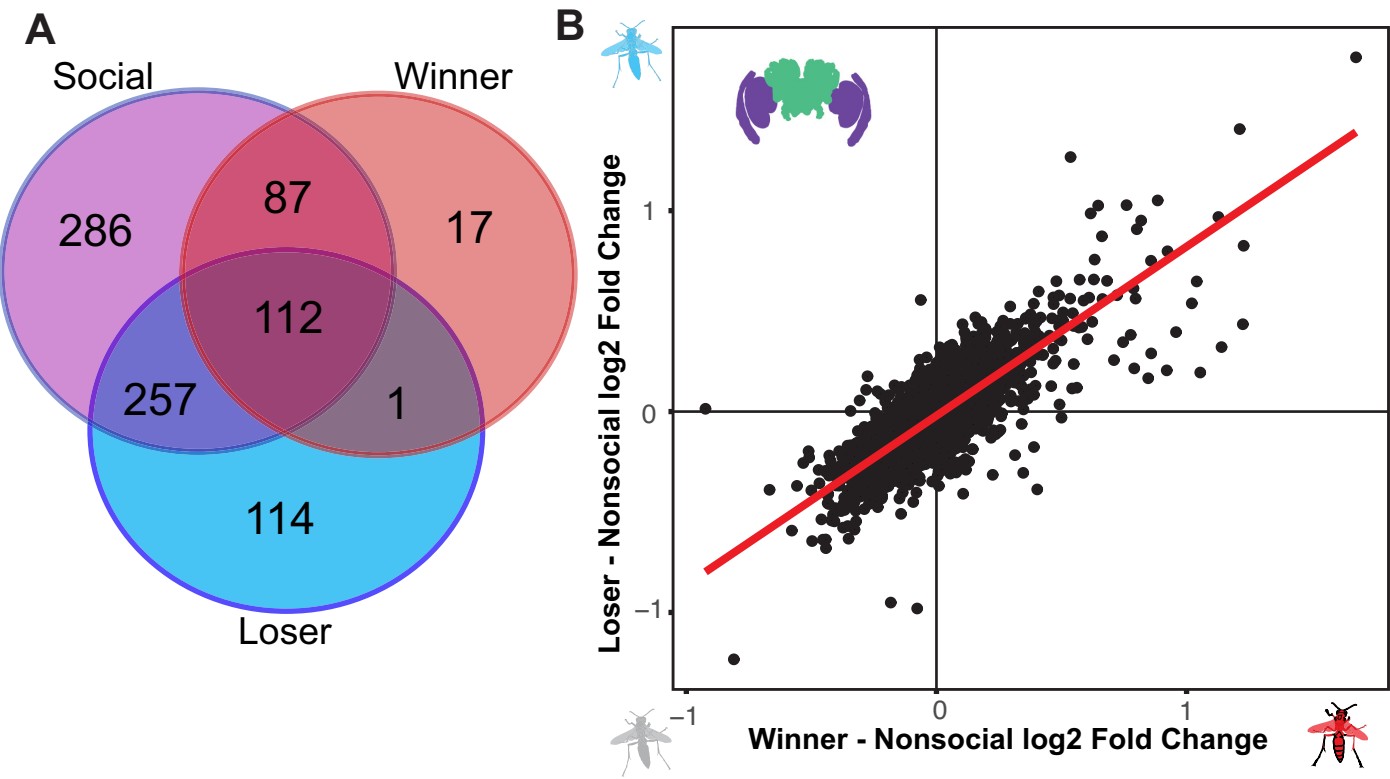

**Fig 4. Similar overall neurogenomic responses in winners and losers.** (A) There is significantly more overlap than expected by chance between the DEGs for winners and losers compared to each other as well as both winners and losers compared to all individuals with recent social experience (P < 2e-16). Note, the circle sizes are not to scale. The numbers within each section indicate the number of genes shared in each area of overlap. (B) The difference in log2 fold change in gene expression for all genes with a mean normalized expression count of 100 or greater for nonsocial individuals are correlated for winners and losers. Both panels show analyses from the entire dataset with both brain regions and time points combined.

criteria for calling DEGs, only 55 genes have P < 0.01 when not correcting for false discovery rates. The lack of strong differential expression between winners and losers suggests that the two social outcomes have similar expression profiles when analyzing the entire dataset, including both brain regions and timepoints. Indeed, there are 113 DEGs shared between winners and losers, a significantly greater overlap than expected by chance (Fig 4A, P < 2e-16). Both winners and losers also show significant overlap with the DEGs responding to social interactions in general (P < 2e-16 in both cases). Next, we compared the patterns of differential expression of winners and loser in relation to the nonsocial wasps. The log2 fold changes in both winners and losers compared to nonsocial wasps in the entire dataset are strongly correlated (Fig 4B, linear model: y = 0.84x - 0.02, $F_{1,4002}$ = 7458, $r^2$ = 0.65, P < 2e-16). Thus, when considering both brain regions and sampling points, winners and losers have broadly similar responses, though with a greater number of DEGs in losers compared to the nonsocial individuals (Fig 4).

We investigated the relationship between gene expression patterns in winners and losers further by comparing the patterns of differential expression relative to nonsocial individuals at the end of the 45-minute interaction and 4 hours later. Here we present the results of gene expression in the brain since we observed stronger effects of social behavior in the brain than optic lobe (S3 Table). We examined the log2 fold change in expression in losers relative to nonsocial individuals in a mixed model with winner log2 fold change relative to nonsocial individuals and time as fixed effects and gene as a random effect. Differential expression between

winners and nonsocial wasps predicts expression differences in losers relative to nonsocial wasps (t = 69.02, df = 7420, P < 2e-16). Time is a significant predictor with greater log2 fold changes in losers compared to nonsocial wasps at the later time point (t = 12.27, df = 3313, P < 2e-16). There is a significant interaction between the extent of differential expression between winners and nonsocial wasps and time (t = 3.3, df = 5424, P = 0.00096). Next, we calculated a separate regression between loser and winner responses compared to nonsocial individuals at early and later times to further investigate these patterns. The slope of the regression is steeper though the fit substantially poorer between winners and losers at the later timepoint compared to the earlier sampling time (Fig 5A, early: y = 0.69x + 0.001, $r^2$ = 0.70; Fig 5B, later:

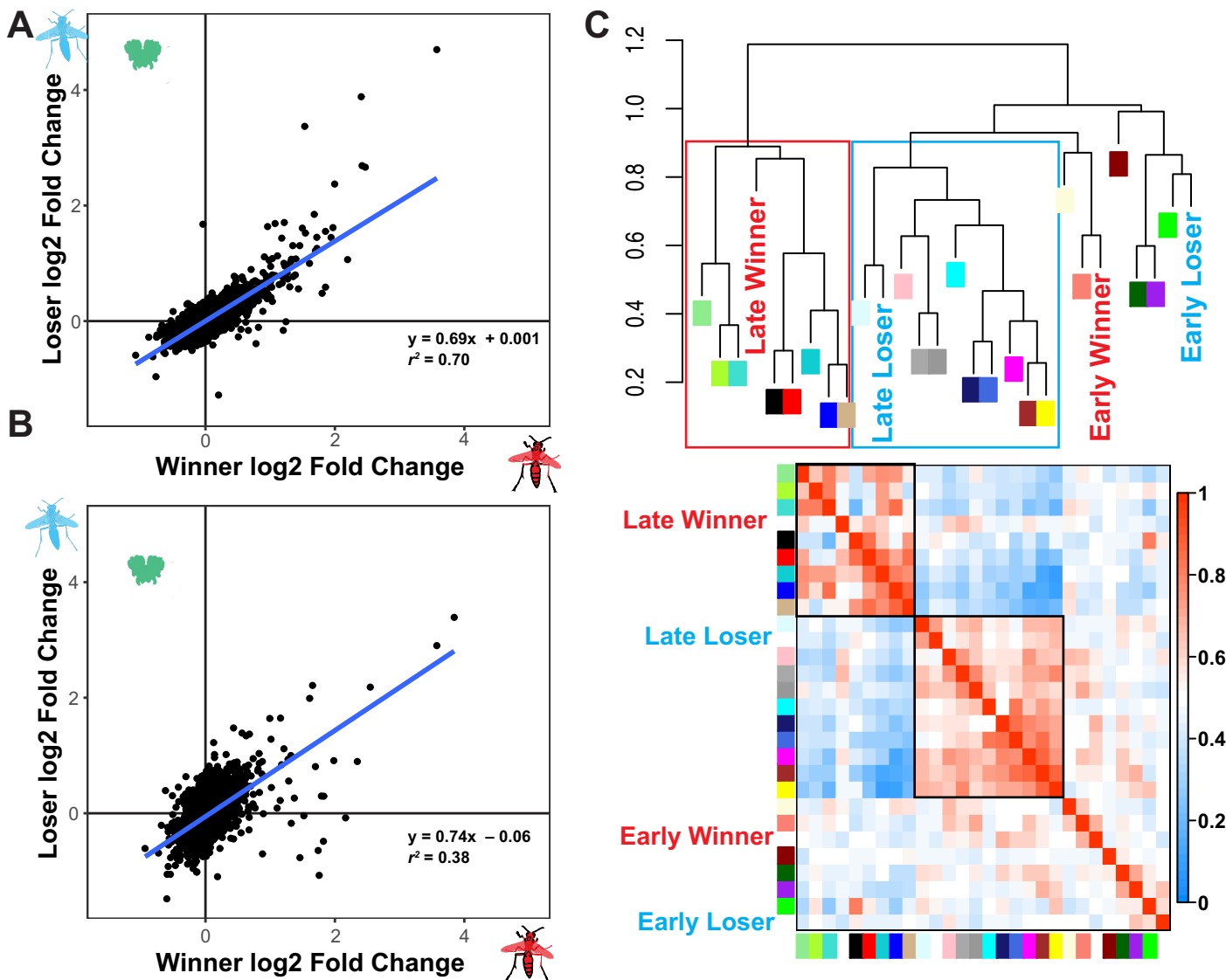

**Fig 5. Greater divergence in loser brain transcriptomes over time.** (A) Focusing on only the central brain dataset, the log2 fold change in gene expression differences between nonsocial individuals and winners and losers are well correlated at the earlier time point. (B) At the later time point, there is substantially less correlation between winner and loser responses relative to nonsocial individuals. (C) Gene correlation modules are organized into two meta-modules, which are associated with late winners and late losers respectively. The top panel shows a dendrogram with the colors labeled and social outcomes labeled. The boxes have been added to highlight the two meta-modules. The bottom panel shows a heatmap showing the relationships among modules. Higher correlations are show by warmer red colors with modules with low or not correlations shown in blue. The two meta-modules highlighted in the dendrogram have been highlighted here with black outlines.

y = 0.74x - 0.06, $r^2$ = 0.38). These analyses suggest the initial similarity in expression between winners and losers declines over time.

Winners and losers show a pattern of increased divergence in brain gene expression over time using a distinct analysis method as well. We used weighted gene-correlation network analysis (WGCNA) to examine patterns of co-expressed genes in relation to social behavior in the dataset of aggressive trials and nonsocial wasps focusing on the brain [53]. WGCNA assigns 6086 genes to 24 modules (S2 Fig, mean = 253.58 genes, max = 1091, min = 39). Multiple modules are significantly associated with winning or losing an encounter. Co-expression modules associated with winning or losing at either time point are all distinct–i.e., no modules are correlated with more than one outcome-time combination (S2 Fig). We examined the relationship among modules and social behaviors by identifying meta-modules, correlated groups of eigengenes, and examining their relationship with different social outcomes. The brain dataset contains two large meta-modules that are associated with late winners and late losers respectively (Fig 5C). In contrast, early sampled losers and winners do not group within clear meta-modules. WGCNA calculates modules blind to the sample attributes such as time of sampling, whether wasps had been given a social experience, or the outcome of that interaction. Nevertheless, WGCNA identifies two distinct gene co-expression meta-modules associated with late-sampled losers and winners respectively reinforcing the observation that antagonistic social interactions lead to increased divergence in neurogenomic states over time.

## Dynamic changes in gene expression in the hours following a social interaction depends on dominance outcome

To investigate the neurogenomic changes that may accompany shifts associated with winning or losing, we compared the relative magnitude of brain gene expression changes between early and late losers to those seen between early and late winners (Fig 6). This analysis follows on the results of Fig 5 and only includes the brain samples and not the optic lobe, where we found little effect of social outcome. There is a statistically significant but very weak negative relationship between the relative changes seen in winners compared to losers (Fig 6: $F_{2,3709}$ = 23.39, P = 1.55e-12, $r^2$ = 0.014). Consistent with the previous analyses (Fig 5), we find that there are more extreme changes in losers compared to winners, shown by the greater spread along the y-axis (Fig 6).

We next examined the identity of genes with the most extreme changes in both winners and losers to learn more about the nature of neurogenomic changes in the brain. Notable genes are highlighted in Fig 6 with additional information provided in S5 Table. We observe multiple patterns of change including genes that are initially upregulated in losers relative to nonsocial wasps at the early time point and then substantially decreased at the later time point. Many of the genes with the largest decreases in losers at the later time show this up-then-down pattern, including *vitellogenin*, *apolipophorin III*, *esterase E4* and *apideacin*. Both *vitellogenin* and *esterase E4* are consistently downregulated in workers compared to queens across Polistine wasps [54]. Comparisons between worker and gyne *P. metricus* found lower levels of *apolipophorin III* in worker- compared to gyne-destined larvae [55]. In *P. canadensis*, workers have increased apolipophorin compared to queens [56]. The gene is also upregulated during usurpation attempts in the socially parasitic *P. sulcifer*, suggesting that gene may have links to aggression in *Polistes* [57]. Apidaecin is an antimicrobial peptide involved in immunity [58] and shows markedly increased expression in losers following social interactions with a later decreases (Fig 6B), suggesting possible immune activation in response to receiving aggression.

We observed multiple genes that show increases in expression over time in losers in the brain. The most upregulated gene in terms of log2 fold change in losers is a *myosin heavy*

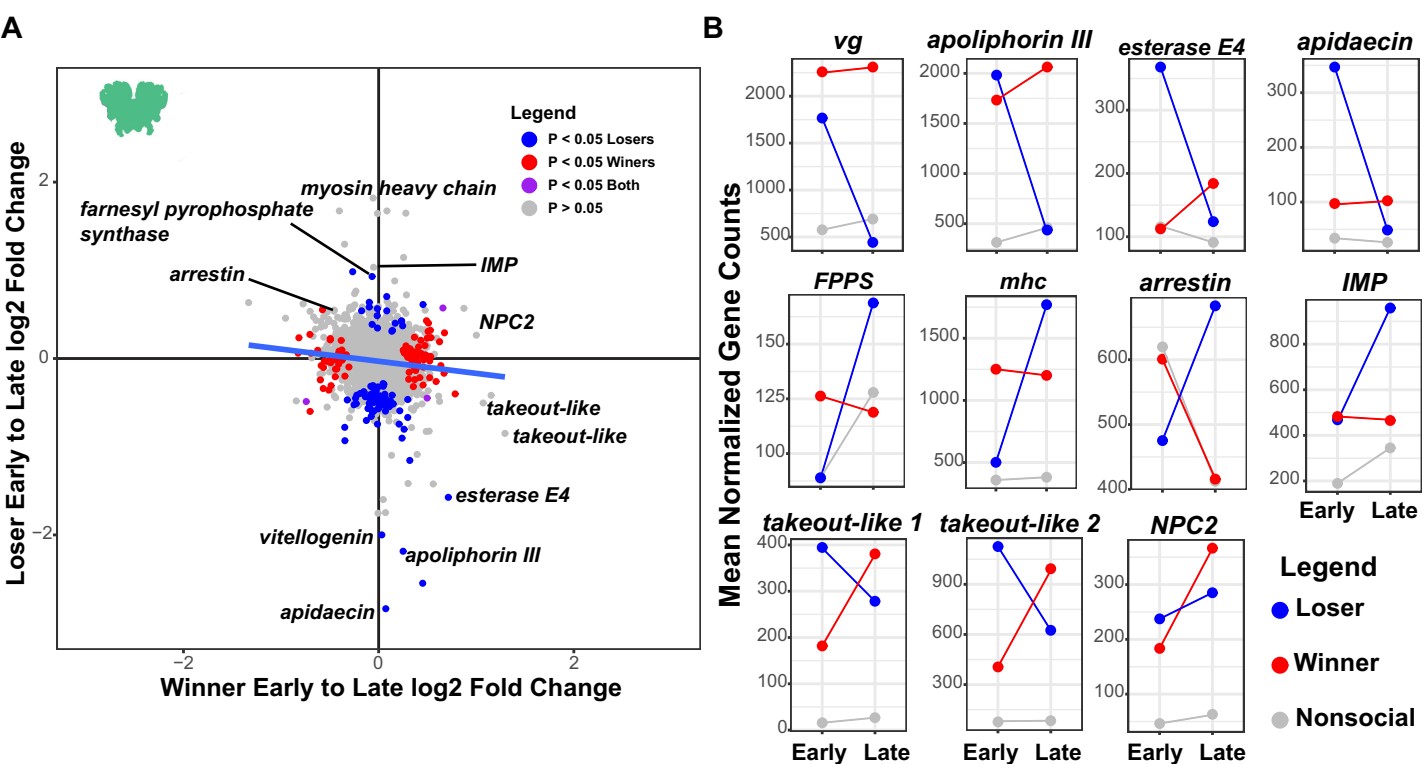

**Fig 6. Shifts in winner and loser gene expression over time.** (A) There are more dramatic shifts in the responses of losers compared to winners over time. The scatter plot shows the log2 fold change between early and late winners on the x-axis against the similar early to late comparison for losers on the y-axis. Thus, genes in the upper right quadrant are those that increase over time in both winners and losers, while those the upper left quadrant increase in losers but decrease in winners. The greater spread along the y- compared to x-axis shows that there are larger changes in loser gene expression profiles over time compared to winners. There is a weak but significant negative correlation suggesting that some genes that increase in losers tend to decrease in winner and vice versa. Notable genes are highlighted. Data points are color-coded according to the legend. (B) The panels show the mean normalized count of expression for losers, winners and nonsocial individuals at early and late sampling points. Lines are drawn connecting the points between groups of the same social outcome. Note that the y-axis scale is different for each gene and depends on the dynamic range of the specific gene. For example, *arrestin* shows a much smaller change in expression across groups than *takeout-like 1*, which is expressed at very low levels in nonsocial controls but expressed much more highly in wasps that engaged in social interactions.

*chain* gene, which are upregulated in social wasp worker brains compared to queens [54]. We also observed a pattern of upregulation of *arrestin* in late losers but down regulation in winners and control nonsocial wasps. Previous studies of caste differential expression in *P. canadensis* found that *arrestin* was upregulated in workers relative to queens [56], and it is found upregulated among foragers in ants as well [59]. *FPPS* encodes farnesyl pyrophosphate synthase which is involved in JH production [60, 61] and is upregulated in queens in Polistine wasps [54]. We also observed increases in *inositol monophosphatase* (*imp*), which is involved in the inositol phosphate signaling pathway [62] and has been linked to task differentiation in ants and bees [63, 64].

Comparing winners at the early and late timepoints revealed less extreme changes than what we observed when comparing losers at the early and late timepoints (Fig 6). Among the genes with largest change by magnitude in winners are two members of *takeout* gene family, which show substantial decreases in losers (Fig 6). The *takeout* gene family is found across insects [65] and are they frequently regulated by juvenile hormone [66–68]. Both winners and losers showed increases in *Nieman Pick Type C2 (NPC2)*, which regulate steroid hormone biosynthesis including juvenile hormone [69], and has been implicated in social communication among ant workers [70]. Notably, all three of these genes are among the most highly and significantly upregulated genes in the brain in response to social interactions (Figs 3D and 6B).

The significant upregulation of these genes in response to social interactions and divergent patterns of expression between winners and losers over time make them interesting candidates for further study.

## Discussion

These data add to a growing body of literature documenting the changes in brain transcriptomic profiles in response to social behavior [2, 5, 8, 11, 16, 24]. The behavioral paradigm used in this study mirrors other lab studies of social behavior and cognition in *P. fuscatus* that examined encounters in a neutral arena and detect variable amounts of aggression [41, 44, 48], though is likely to be a less extreme social experience compared to paradigms that challenge individuals in their nest or home cage and or otherwise produce strong fighting responses used in other behavioral transcriptomic studies [5, 8, 10, 11]. Although the social experiences in our trials were comparatively mild, we nevertheless detect hundreds of differentially expressed genes in response to social interactions. The neurogenomic effects of social interaction are detectable at both the earlier (at the end of a 45-minute interaction) and later (4 hours following the interaction) time points, but the evidence for differential gene expression between social and nonsocial individuals is strongest shortly following an interaction (Fig 3C). The transcriptomic signatures measured right after the interaction represent a combination of immediate responses to social stimuli and interactions as well as some of the initial downstream physiological responses to social behavior. In contrast, at the 4 hour timepoint individuals had been removed from social interactions for a period of time so socially regulated genes at this later timepoint likely reflect downstream consequence of social interactions [3]. The increased number of differentially expressed genes at the earlier timepoint may reflect the engagement of a broad set of neural circuits and gene-networks during social interactions. Conversely, the decrease in differential expression over time is driven at least in part by the divergent response to social outcomes from winners and losers, such that there is more diversity in the transcriptomic signatures of the wasps a few hours after the interactions.

We detected stronger effects of social experience on the brain compared to the optic lobe, though we did detect some effects of social experience on optic lobes transcriptomes. There is a growing literature demonstrating that sensory system tuning and function is more dynamic and plastic than has been previously appreciated [71–76]. Though examples of sensory plasticity are often developmental shifts in response to predictable cues such as season or reproductive state, there is also evidence that individuals' sensory systems respond to their physical environment [76, 77]. We examined the responses of optic lobes to social interactions in paper wasps and found modest evidence of differential expression 4 hours after social interactions (Fig 3C). Among the differentially expressed genes include a dopamine transporter and a major royal jelly protein, which are both downregulated in the 4-hour time point in social compared to nonsocial wasps, suggesting the possibility for modulatory effects on the visual system following social interactions. It is possible that longer-term exposure to social interaction or isolation could have more dramatic effects on visual systems. Indeed, social experience during development is required for individual recognition in *P. fuscatus* [78] and normal development of higher order visual processing regions of the wasp brain [47].

### Transcriptomic signatures of memory

Paper wasps form stable long-term memories of individuals they have interacted with, so we were interested to look for signatures of differential expression associated with memory. Previous transcriptomic studies of memory using associative learning paradigms in bumblebees had found evidence for differential expression of memory-related genes after a few hours [45].

Our study also detected an enrichment of differential expression of genes with GO annotations related to memory. A puzzling feature of our findings, however, is that the differential expression was not in a consistent direction. Studies of the genetic basis of memory formation in invertebrates have often focused on single cue associations (e.g., a color or smell) [45] but the social interactions studied here are more complex in terms of sensory inputs and the range of positive and negative experiences that occur. Global downregulation in the brain may mask upregulation in specific neurons where social memories are encoded. We detected enrichment in both anesthesia-resistant and long-term memory annotated genes. Anesthesia-resistant memory refers to a process of memory consolidation that is resistant to disruptions in neural activity, as would be caused by anesthesia [79]. Anesthesia-resistant memory does not require protein synthesis and is considered a form of intermediate-term memory [80, 81]. Long-term memory in contrast requires protein synthesis and the reweighting of synaptic connections [82, 83]. Long-term memory formation requires modulation of synaptic connections [52], which may be captured by GO terms dealing with changes to synapses including their plasticity and targeting. The two time points in our study, then, may be capturing dynamic changes in the types of memories that are formed as intermediate-term memories may give way to long-term memories as time passes. Though all the individuals were sacrificed at one of two pre-determined timepoints relative to the start of their trials, the timing and intensity of interactions were not uniform across the subjects, which likely have some influence on the transcriptional dynamics of memory for each individual. Additionally, enrichment for GO terms related to mushroom body development when seen in the context of an adult brain, are suggestive of a role of mushroom body neuropils in social processing and memory. The context or features of an interaction that make it more or less memorable for paper wasps remain to be investigated, though the present study was able to detect neurogenomic signatures related to memory following interactions in a neutral arena. How investment in memory may vary across social contexts (on a nest versus a neutral arena) and the intensity of the interactions are open questions that the present data suggest could be addressed, at least in part, using transcriptomic techniques. While these data demonstrate that social interactions influence the expression of memory-related genes, understanding how these patterns translate to memory formation (or lack thereof) will require further study.

## Social outcomes cause later divergence in brain transcriptomes

We observed initially similar transcriptomic responses between winners and losers that diverged over time. The overall neurogenomic responses to social interactions were similar in winners and losers observed in the whole dataset (Fig 4) was driven by their initial similarity at the end of the interaction (Fig 5). The responses diverged over the course of a few hours, with relatively greater differences in losers over time relative to individuals that did not have a social interaction. The correlation between winners and losers at the early time point echoes shared patterns of neural activity observed in mice and bats or shared transcriptomic signatures among interacting individuals in beta fish [5–7]. Given that competitive social interactions typically lead to divergent outcomes for winners and losers or dominants and subordinates [4, 18, 22, 84, 85], the initial similarity in neural responses between competing individuals may seem counterintuitive. The similar neurogenomic responses of winners and losers observed at the earlier timepoint, however, declines over time in our dataset. The similar early responses may reflect the activity of neural mechanisms for assessing social stimuli and the initial processing of the encounter that is shared between the interacting individuals. Divergence over time may reflect the integration of the outcome into neurogenomic responses that themselves go on to further influence behavioral states following social encounters. This divergence

among socially interacting wasps likely contributed to the reduced number of differentially expressed genes detected between social and nonsocial treatments at the late time point due to heterogeneity in expression patterns between winners and losers. Reproductive division of labor among groups of foundresses is based on physical aggression in *Polistes* [28, 29], but ultimately results in distinct neural and physiological states between the dominant and subordinate foundresses [38, 86]. Understanding the steps that lead from similar to divergent neurogenomic states between interacting individuals will help clarify how social experiences come to generate diversity in physiology and behavior among individuals in a population [3, 87].

It is notable that we found more differentially expressed genes in losers than in winners. This observation fits with theoretical results that loser-effects should be stronger than winner-effects [88]. Losers in our experiment would potentially become subordinate foundresses in a natural nesting context and not workers, though subordinates do more foraging than dominants [31]. Despite reduced reproduction and greater foraging relative to dominant foundresses, subordinate foundresses are not the same as workers and have been shown to have distinct neurogenomic profiles compared to dominant foundresses and workers in microarray and candidate-gene studies [38, 86, 89, 90]. Nevertheless, the expression patterns of these genes suggest that within a few hours of emerging from a social encounter as a subordinate, multiple genes are dynamically regulated in a manner suggesting changes to aggression, reproduction, and metabolism (Fig 6).

There are a number of interesting candidate genes that are highlighted by their level of dynamic change following social interactions including some that have previously been associated with caste differences (e.g., *myosin heavy chain*) as well as genes that have not previously associated with aggression or caste (e.g., *takeout-like* genes) (Fig 6). Among the genes identified, however, one stands out due to its long history in social insect research as well as its notably different dynamics in winners, losers, and nonsocial wasps–*vitellogenin*. Vitellogenin (*vg*) is classically recognized for its role as an egg-yolk protein, which has a conserved role in oogenesis across insects [91]. In paper wasps, levels of *vg* in the head or brain have been associated with social status, being highest in single and dominant foundresses and lowest in subordinate foundresses and workers [38, 54, 56, 89]. Our data suggest that *vg* levels quickly respond to social interaction, rising substantially in both losers and winners relative to nonsocial controls at the early time point (Fig 6B). Winners maintain high levels of *vg* for hours after the interactions, while levels plummet in losers below those seen in nonsocial controls. Nonsocial control wasps show relatively lower levels of *vg* compared to socially interacting wasps, though it is hard to contextualize the *vg* levels observed in control wasps compared to those reported in other studies. Previous studies have examined patterns of gene expression in wasps in relation to life history state or broader social contexts (e.g., foundresses versus worker) and not in response to specific social experiences [38, 54, 56, 89]. Additionally, the wasps in this study had been kept in the lab without nests prior to the trials, following other studies of staged aggression contests [35, 36, 44], which likely influences baseline levels of gene expression. Nevertheless, we found that *vg* is strongly upregulated in response to social interactions in general, but expression levels then diverge depending on social outcomes. To the extent that *vitellogenin* influences levels of aggression, the decrease seen over time in losers in this study may be indicative of a shift toward a submissive behavioral state.

## Conclusions

The analysis of 139 RNAseq samples from the optic lobes and brains of *P. fuscatus* foundresses revealed novel insights into the dynamic changes in neurogenomic states following social

interactions. Female *P. fuscatus* paper wasps have variable facial patterns that they use to visually recognize each other as individuals [36, 40]. Though we did detect some differentially expressed genes in the optic lobe transcriptome in response to social interactions, changes in the brain were much larger and more dynamic, likely reflecting the importance of processing and integrating socially relevant information as a key factor in driving neurogenomic shifts. After a 45-minute interaction, winners and losers show similar average changes in patterns of gene expression relative to nonsocial individuals, which may reflect the fact that the same neural circuits likely process initial social interactions regardless of the outcome. This result mirrors recent findings of similar neural firing patterns during social interactions in rodents and bats [6, 7] and similar neurogenomic responses shortly after fights in beta fish [5].

Over a span of 4 hours the initial similarity between winners and losers decreases, as loser gene expression patterns show larger shifts consistent with theoretical predictions of larger loser effects compared to winner effects [88]. The most dramatic shifts in expression over the course of four hours in losers are due to a mixture of increasing or decreasing expression time (Fig 6B). These data suggest that within a few hours a single subordinate experience can influence expression of multiple genes associated with behavioral and physiological differences, perhaps most notably *vitellogenin*. We do not suggest that a single social experience is necessarily sufficient to make a wasp into a subordinate foundress. Paper wasps engage in aggressive interactions on and off the nest early in the nesting cycle [31] and many wasps that go on to become solitary or dominant foundresses likely experience some social defeats during this phase. Repeated interactions between co-nesting foundresses, however, may compound and reinforce the types of effects we observe. Neurogenomic studies show shifts in neurogenomic profiles in many caste-associated genes in response to repeated wins or losses in dominance contests in *Ceratina* carpenter bees [16, 24]. Paper wasps are notably plastic, with aggressive and dominant workers becoming more queenlike in the span of a few days when reproductive opportunities become available through experimental removal of the queen [26, 27]. Moving forward, a major challenge is to understand how social experiences are processed in the brain giving rise to neurogenomic shifts and changes in expression of key regulators of behavior such as *vitellogenin*. Specifically, single-cell RNAseq approaches have the potential to indicate which cell-types are most strongly influenced by social interaction and could reveal how diverging gene expression patterns give rise to broader physiological consequences associated with social status.

## Methods

### Experimental design and behavioral scoring

We tested the role of social experience on neurogenomic states comparing the responses of individuals to staged contests in a neutral arena to solitary experiences in the same arenas. Subjects were 90 female *P. fuscatus* foundresses collected during the pre-worker colony phase from their nests or while foraging in Tompkins County, New York in the spring of 2018 (S6 Table). All of the spring-caught foundresses would have emerged as adults at the end of the previous summer (approx. Aug-Sep 2017). Details of collection for each wasp are provided in S6 Table. Wasps were brought into the lab and provided housing in small deli cups with *ad libitum* access to sugar and water. Prior to the trials, wasps were given identifying paint marks using Testor's enamel paint to facilitate scoring of social interactions. During the trials, wasps were placed in a small neutral arena (100 mm diameter clear petri dish) with a plexiglass-lid under bright full spectrum lights either alone or with another wasp. Social trials featured pairings between weight-matched wasps that had been collected at distinct locations at least 2 kilometers apart, which is greater than the typical dispersal distances for this population [92].

While in the arenas, wasps were filmed for 45 minutes and then removed from the arenas. In half of the trials, wasps were immediately sacrificed by decapitation and their heads were placed in RNAlater for subsequent analysis. To aid uptake of RNAlater, small cuts were made on the exoskeleton of the head avoiding damage to neural tissue. In the other half of the trials, the wasps were returned to their individual housing and sacrificed 4 hours later using the same protocol. This generated four sets of samples: early social wasps (n = 30 wasps from 15 trials), early nonsocial wasps (n = 15), late social wasps (n = 30), and late nonsocial wasps (n = 15, Fig 1).

Videos of the social wasps were scored for stereotyped paper wasp aggressive behaviors including mounting, biting, hitting, grappling and darting [31, 41]. Additionally, we scored when one wasp chased the other as an aggressive act. Following previous research we calculated an aggression index to capture the relative intensity of fighting [44, 48]. We gave each act a weight aggression score from 1 to 5, with 5 indicating higher intensity, using the following rubric: dart = 1; snap = 2; chase, hit, lunge = 3; grapple, bite = 4; mount = 5. Summed aggression scores are simply the summed score of all actions. The aggression index is the average aggression score. We compared the rates of aggression between individuals in the trials using linear models. All of the behavioral data is provided as S1–S3 Data.

## RNA sequencing and read processing

Brains and optic lobes were dissected from RNAlater-preserved wasp heads under a stereomicroscope. Optic lobes were separated from the rest of the brain (Fig 1) and then the two optic lobes (right and left side) were combined for processing while the remainder of the brain was processed as a separate sample. We refer to these two tissue segments simply as the optic lobes and brain respectively in the text. RNA was extracted separately from the brain and combined optic lobes generating two pools of RNA from each wasp. Extracted RNA samples were sent to the Cornell Genomics Core for 3'RNA library preparation using the Lexogen kit. Due to low and/or poor-quality RNA yields for some samples, we were able to sequence 168 samples out of the intended 180. We sequenced libraries using single end 50 bp reads on a NextSeq500. In 3'RNAseq each transcript is only covered with a single read meaning that the read number is equivalent to total number of transcript counts sampled, reducing the total necessary read depth because the entire length of the transcript is not sequenced.

We mapped reads to the *P. fuscatus* genome [42] using STAR [93] with the following command: STAR—genomeDir ~/STARgenome—runThreadN 30—readFilesIn 'filename_here'. fastq.gz—readFilesCommand zcat—outFileNamePrefix filename_here'_—outFilterMultimapNmax 1—outReadsUnmapped unmappe_a—outSAMtype BAM SortedByCoordinate. Read counts were calculated using HTseq with default settings [94]. Initial read counts revealed that the annotation of the *P. fuscatus* genome did not capture many 3' untranslated regions, so we manually scanned the genome to update gene body annotations. To identify 3' untranslated regions we jointly visualized paired-end mRNAseq reads from female *P. fuscatus* heads with a sample of 3' RNAseq reads using the Integrated Genome Viewer [95] and updated a GTF file based on this scan. In addition to extending the UTRs, in some cases we combined genes, separated genes or identified genes not previously included in the prior annotation. The GTF file used for this study is provided as a supplemental file (S4 Data). We removed samples with less than 1 million gene counts, since they had been sequenced much less than the other samples. In the end, samples had an average of 5.15 million unambiguous gene counts (range 2.05–12.14 million gene counts, S7 Table). We removed non-expressed or very lowly-expressed genes from the count table in order to make analyses faster. After filtering, we were left with 8219 genes for further analyses. Before engaging in downstream

differential expression analyses, we first inspected the separation of the samples using principal component analysis (PCA) to ensure that brain and optic lobe tissues had distinct expression profiles, as would be expected based on differential cellular composition of the samples. The PCA was calculated by using the 'vst' normalization function of DESeq2 [96]. Inspection of the samples plotted against PC1 and PC2 revealed 2 distinct clusters of samples corresponding to optic lobe and brain respectively (Fig 3A).

## Gene expression analyses

Patterns of differential expression were determined using DESeq2 [96] in R v 3.6.2 [97], because the program performs well when there are relatively small number of samples for each conditions as is the case for our study. Input files used for gene expression analyses are given in S5–S6 Data. Depending on the analysis we examined the entire data set (both brain and optic lobe), only the brain data or only the optic lobe data using linear models with fixed effects. All R code used for analysis is provided as a supplemental file (S1 Text). First, we considered models with social experience treatment (social v. nonsocial), brain region (brain v. optic lobe) and time (early v. late). We examined the interactive effects following the recommendations of the authors of the DESeq2 analysis package [96]. We generated combined variables to examine differences in expression across groups. For example, to look at the effects of time and social experience we classified samples as belonging to one of four groups early_social, early_nonsocial, late_social, or late_nonsocial under a single categorical variable, i.e., time_social. By comparing contrasts among the different pairs of categories, we were able to determine how different combinations of samples influence patterns of differential gene expression. In analyses of the effects of social interaction per se, rather than outcome, we analyzed all RNA samples. For analyses looking at contest outcome, we only examined social trials for which at least 10 aggressive acts occurred. Removing the least aggressive trials results in detecting more differentially expressed genes despite having a smaller sample, suggesting that the least aggressive trials add noise to the dataset. The outcome of the trial was coded as winner, loser or nonsocial. Genes were considered to be differentially expressed if the FDR adjusted P value ≤ 0.1. We did not use a logFold cutoff for our analyses as our goal is to understand the broad changes in expression profile and not merely identify the most differentially expressed genes; applying such a filter would unnecessarily lose information [97]. The same gene count datasets were used for PCA, DESeq2 and WGCNA analyses. In some subsequent analyses of log2Fold change patterns from DESeq2 output, however, additional filtering was applied to remove lowly expressed genes (see below).

   We compared patterns of differential expression in winners and losers, based on log2 fold changes in expression. Since absolutely small changes in lowly expressed genes can give rise to large log2 fold changes, we removed all genes with mean normalized expression counts below 100 before comparing patterns of expression across time points and outcomes. Note, this filtering was done on the output from DESeq2, not the input. Thus, the analyses examining the relationship between winner and loser expression based on DESeq2 output only examine more highly expressed genes. First, we compared expression relative to nonsocial wasps in winners and losers respectively in the combined brain and optic lobe datasets. In analyses focusing solely on the brain dataset, we examined how the relations between winner and loser expression profiles changed between early and late sampling points using a general linear mixed model implemented in the lme4 package for R [98]. Log2 fold change differences in expression relative to nonsocial wasps sacrificed at the same time were used as a basis of comparison. We modeled relative fold change in losers as a function of the relative fold change in winners, time, their interaction, and gene ID as a random effect. We also separately examined the

relationship between winners and losers at early and late time points using a linear model. Finally, we compared the relative log2 fold changes between the earlier and later time points for losers to the changes observed in winners. In these comparisons positive values of expression denote increased expression at the later time point.

Genes are frequently expressed in a modular manner, with groups of genes showing similar expression patterns [53], so we calculated co-expression modules from our brain dataset using WGCNA. This analysis focused on understanding modules associated with winning or losing at different time points, so we limited our analysis to a subset of the brain RNAseq data set that had engaged in more vigorous encounters (i.e., winner, loser and nonsocial). We report Pearson correlations. We used a soft threshold power of 10, determined from visual inspection of a power analysis. Our network used signed correlations. R code used for analysis is provided as S1 Text.

### Gene ontology

We used gene ontology enrichment analyses to identify gene functions that were enriched in our dataset of socially regulated genes. The *P. fuscatus* gene set was annotated using the Blast2GO function of OmicsBox based on sequence similarity with *Drosophila melanogaster* genes [99]. For enrichment analyses, we used the TopGO package in R [100]. We only included categories with at least 10 annotated genes in the dataset. Significantly over-represented categories were identified using the 'weight01' function in TopGO with the 'classicfisher' statistic. The 'weight01' function returns a P-value conditioned on other neighboring terms in the gene ontology hierarchy [101]. We were specifically interested in assessing the *a priori* hypothesis that social interactions would have a transcriptomic signature related to memory based on previously behavioral work and so report the P values for the Fisher's exact tests from TopGo in the main text. However, enrichment analyses in TopGo by default test all the possible GO terms, which raises issues regarding multiple testing. We provide a list of all 3096 gene ontologies output from TopGO results along with a FDR calculated using the 'fdr' method with the 'p.adjust' function in R so readers can assess the pattern of enrichment (S4 Table). After FDR correction, 5 GO terms show significant enrichment (FDR < 0.1) including anesthesia resistant memory (GO:0007615, FDR corrected P-value = 0.02).

### Supporting information

**S1 Fig. Volcano plot of socially regulated genes in full dataset.** The volcano plots show the genes that are differentially expressed between individuals that experiences social interaction (social) versus those that did not (nonsocial). Higher log2 fold change values indicate upregulation in the social group compared to the nonsocial group. Panel A shows the data for all genes examined. Panel B shows the plot for genes annotated with memory-related functions. The red and blue striped wasp symbol indicates that the data includes all socially interacting wasps.
(PNG)

**S2 Fig. Module outcome relationships in the brain dataset.** The heatmap shows the relationships between the modules of co-expressed genes identified by WGCNA with different social outcomes. White and lighter colors indicate low correlations. Bolder reds indicate stronger positive correlations. Bolder blues indicate stronger negative correlations. In each cell the top line of text reports the correlation between a given module and the social trait or time since social interaction. The bottom line of text shows the P value in parentheses. Significant

correlations (P ≤ 0.05) are highlighted with bold text.
(PNG)

**S1 Table. Behavioral data.**
(XLSX)

**S2 Table. PCA output.**
(XLSX)

**S3 Table. DEGs from different models for brain and optic lobe data.**
(XLSX)

**S4 Table. GO term enrichment analysis.**
(XLSX)

**S5 Table. Notable genes from Fig 6.**
(XLSX)

**S6 Table. Collection data for wasps used in the study.**
(XLSX)

**S7 Table. Sequencing data summary.**
(XLSX)

**S1 Data. Raw behavioral data.**
(CSV)

**S2 Data. Behavioral data per trial.**
(CSV)

**S3 Data. Behavioral data per wasp.**
(CSV)

**S4 Data. GTF file used in this study.**
(TXT)

**S5 Data. Metadata file for all samples used as input to DESeq2 and WCGNA analyses.**
(CSV)

**S6 Data. Raw gene counts for all samples used as input to DESeq2 and WCGNA analyses.**
(CSV)

**S1 Text. R code used or analyses.**
(R)

**S1 Movie. Video of a social trial showing two wasps interacting.**
(MOV)

## Author Contributions

**Conceptualization:** Floria M. K. Uy, Christopher M. Jernigan, Michael J. Sheehan.

**Data curation:** Floria M. K. Uy, Michael J. Sheehan.

**Formal analysis:** Michael J. Sheehan.

**Funding acquisition:** Michael J. Sheehan.

**Investigation:** Floria M. K. Uy, Christopher M. Jernigan, Natalie C. Zaba, Eshan Mehrotra, Sara E. Miller.

**Methodology:** Floria M. K. Uy, Christopher M. Jernigan.

**Project administration:** Michael J. Sheehan.

**Resources:** Michael J. Sheehan.

**Supervision:** Michael J. Sheehan.

**Visualization:** Sara E. Miller, Michael J. Sheehan.

**Writing – original draft:** Michael J. Sheehan.

**Writing – review & editing:** Floria M. K. Uy, Christopher M. Jernigan, Natalie C. Zaba, Eshan Mehrotra, Sara E. Miller, Michael J. Sheehan.

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
