## [Decision Letter · Decision Letter 0]

6 Apr 2021

Dear Dr Sheehan,

Thank you very much for submitting your Research Article entitled 'Dynamic neurogenomic responses to social interactions and dominance outcomes in female paper wasps' to PLOS Genetics.

The manuscript was fully evaluated at the editorial level and by independent peer reviewers. The reviewers appreciated the attention to an important problem, but raised some substantial concerns about the current manuscript. Based on the reviews, we will not be able to accept this version of the manuscript, but we would be willing to review a much-revised version. We cannot, of course, promise publication at that time.

If you decide to revise the manuscript for further consideration at PLOS Genetics, please aim to resubmit within the next 60 days, unless it will take extra time to address the concerns of the reviewers, in which case we would appreciate an expected resubmission date by email to plosgenetics@plos.org.

[LINK]

We are sorry that we cannot be more positive about your manuscript at this stage. Please do not hesitate to contact us if you have any concerns or questions.

Yours sincerely,

Lisa Stubbs

Associate Editor

PLOS Genetics

Hua Tang

Section Editor: Natural Variation

PLOS Genetics

Reviewer's Responses to Questions

**Comments to the Authors:**

Reviewer #1: How the outcomes of social interactions affect future behavior through the brain remains an area of great interest to sociobiology. Here, the authors use the primitively eusocial paper wasp *Polistes fuscatus* to examine the effect of a social challenge and the valence of its outcome on the global transcriptional state of the brain during the minutes and hours after these interactions.

The authors take advantage of the interesting biology of these paper wasps, who can recognize individual conspecifics, to differentiate the immediate and later effects of social challenge on the optic lobes and central brain. In general, this is a well-designed and well-powered study. The bioinformatics and statistical techniques used to analyze the RNA-seq data were appropriately chosen and applied. The authors have made their data available and say they have made their code available according to the standards of the genomics field. Overall, this is an excellent addition to the budding field of comparative genomics of social behavior. Figure 5, in which differences manifest in expression between winners and losers, is a particularly important contribution.

My comments are relatively minor and mostly confined to issues of dataset presentation.

* I can’t access the authors’ code. I’m not certain if they’ve uploaded it to the journal website and it can’t be accessed because of limitations with the editorial manager or if they haven’t yet made the code public on a resource such as GitHub. Please make sure the code is accessible on publication.

* In Lines 162-183, the authors state that the PCA does not strongly separate social from non-social samples. However, the results of the PCA are only reported for the first two principle components. Have the authors examined subsequent principle components? What proportion of variance is explained by PCs 1 and 2 versus subsequent PCs?

* Lines 223-234: a false discovery rate correction should be applied to the Gene Ontology enrichment results to account for multiple testing. As they have 61/742 gene sets at a raw p < 0.01, I believe the authors will still find plenty of evidence for overrepresented GO terms after FDR correction is applied. A threshold such as FDR < 0.10, which the authors used for differential expression, would be appropriate.

* The paragraph in lines 316-331 (WGCNA results) contains some verb tense inconsistencies with the paragraphs flanking it. Please revise to use a consistent verb tense.

* Line 549: I’m assuming that the threshold of average expression below 100 means that they filtered out genes with an average of 100 counts per sample or less. Please label units to clarify.

* Lines 563-568: though it should be possible to find this information in the code associated with the paper, please give more details about the parameters used for WGCNA. Specifics that should be included are: which correlation was used (mid-weight bicorrelation or Pearson)? How was the soft threshold established? Were coexpression modules constructed using signed or unsigned correlations? How many genes were in the grey (unclustered) module?

* Figure 3: the Venn diagram panel does not use areas scaled to the size of gene sets in each bin. The authors should re-make this panel to present the gene sets as scaled. A suggested R package is VennDiagram, which has facilities for scaled Venn diagrams (see Figure 1B in PubMed ID: 21269502).

* Figure 4C: it would be helpful if the colors of the bounding boxes around the dendrogram meta-modules and the colors of the bounding boxes around the heat map meta-modules matched.

* A supplementary table containing the gene IDs of all genes highlighted in red, blue, or purple in Figure 5A would be helpful to others interested in this work.

* Figure 5B please label the unit of measurement (CPM, TPM, RPKM?), preferably in the Y axes of the figure panels.

Reviewer #2: In the manuscript “Dynamic neurogenomic responses to social interactions and dominance outcomes in female paper wasps” Uy et al. describe the brain transcriptomics responses to competitive social interactions in female paper wasps, Polistes fuscatus, The authors find that subordinate wasps are less aggressive, forage more, and lay fewer eggs compared to dominant foundresses. They also find considerable differences in brain gene expression, including those related to aggression, worker behavior, and reproduction. Overall, this is an interesting study that will make a valuable contribution. I have a number of comments and suggestions that will hopefully allow the authors to improve the manuscript further.

General comments:

1) Both the Introduction and Results/Discussion are difficult to follow and can be structured more clearly. The authors start out with a big question (“How are social outcomes translated into physiological changes?”), and then meander through different topics in a way that makes the reader’s head spin. It is not until the final paragraph of the Introduction that the specific questions asked in this study become clear. Similarly, the presentation of the Results can be streamlined. Finally, I think the paper would be more compelling if the Discussion would be separate from the Results.

2) I understand the rationale for examining the optic lobes and “central brain,” though given the repeated emphasis on cognitive and similar higher order processing, I wonder whether the authors considered analyzing gene expression in the mushroom bodies.

3) Provide more information on the behavior data. For example, consider showing representative timelines of behavior (one aggressive act every 1.5 minutes seems low?) and graphs that summarize behavior for winners and losers (along with appropriate statistical analysis). This would also help the reader to get a better understanding of what a “more vigorous encounter” (line 565) is as opposed to most trials being “comparatively mild” (line 181; compared to what?), and why only social trials with at least 10 aggressive acts were analyzed (line 543).

4) The presentation of the PCA results can be improved. Consider showing scree plot (to show how variance components break down) and box plots with statistics for PC1 and PC2; providing information on whether higher order PCs separate any of the experimental groups. Because most of the variance is explained by brain region (as expected), I suggest conducting PCA for each brain region separately as well.

5) The authors used DESeq2 for DEG analysis. Given the experimental design, limma might be more appropriate.

Specific comments:

Line 52: Provide a reference for physiological response to cooperative interactions.

- Line 67f: See also the recent study by Vu et al. (https://doi.org/10.1371/journal.pgen.1008831), which is cited later on.

- Lines 282-285: Consider moving these sentences before line 277.

- Line 317: WGCNA stands for weighted gene co-expression network analysis, not “correlation network analysis.”

- Line 319: Provide number of genes for each module (this info could be added to figure 4C)

- Line 477: How long are the wasps in the lab before testing?

- Line 489: Consider rephrasing such that it is clear that each group has the same n = 15, as there are winners and losers for both sets of social wasps.

Line 512: What parameters were used with STAR?

- Line 526f: Can you be more specific on the filter criteria used for removing genes? And why not use the same criteria for both PCA and DEG analysis?

- Line 544: It is common to use a log-fold cutoff, but I don’t think it is necessary in this case. This should be clearly stated, however.

Reviewer #3: This is an interesting study of the brain gene expression in paper wasps as a result of social aggression or social isolation. The time course and replication of this study are well in line with former studies, but the sequencing effort and data analysis presented here lack comparisons with other relevant literature.

Results and Discussion

Lines 162-164: No mention of experimental design or behavioral results are presented here making it difficult to understand sample sizes, focal groups or behavioral assays. Some inclusion of the sample sizes, experimental design, and behavioral results are needed here since this information is curiously omitted from the results.

LIne 171-172: here again the reader has no idea what social and non-social mean in this context. Some behavioral results above would improve the readability.

Lines 176-178: again the reader has no ability to assess this and it would be more useful to explain the behavioral treatments and design for this work then to state they are similar to other studies. Expand on this to explain the number of wasps and numbers of assays.

Line 214: needs more justification why optics lobes were dissected in this study versus rest of brain. Why optic lobes and not antennal lobes or mushroom bodies?

Methods

Line 476: How many wasps per assay amount to the 90 focal females? Were all nests from different nests or how was this handled? More specifics on the age, reproductive status and colony of origin are needed here.

Lines 493-498: This data should be included in the results. Currently there is no mention of aggressive behavior or statistical treatment of assays to follow how winner and losers and time course data were generated.

Lines 507-510: sample sizes are explained here but not by treatment or time point. How did the loss of samples from 180 to 168 to 139 reflect in social vs non-social and 45 min vs 4 hour treatments. What are the sample sizes per treatment.

Line 509: 1 milion reads per sample seems very low. What is the justification for this? Most studies use 30-60M reads per sample for RNAseq studies of this kind. Viewing Table S1 only 2-5M reads were used for this study which is an order of magnitude less than comparable work. What effect does this have on the results presented here?

On a related note, there is no explicit comparison of these results to any other studies in honey bee, mice or sticklebacks, etc. Is this because of the low sequencing coverage or what were hypergeometric test comparing gene lists from other studies not explored here? Are the genes identified here significantly similar to other studies?

**Have all data underlying the figures and results presented in the manuscript been provided?**

Reviewer #1: Yes

Reviewer #2: Yes

Reviewer #3: Yes

PLOS authors have the option to publish the peer review history of their article (what does this mean?). If published, this will include your full peer review and any attached files.

Reviewer #1: **Yes: **Michael C. Saul

Reviewer #2: No

Reviewer #3: No

---

## [Decision Letter · Decision Letter 1]

14 Jul 2021

Dear Dr Sheehan,

Thank you very much for submitting your Research Article entitled 'Dynamic neurogenomic responses to social interactions and dominance outcomes in female paper wasps' to PLOS Genetics.

The manuscript was fully evaluated at the editorial level and by independent peer reviewers. The reviewers appreciated the attention to an important topic but identified some concerns that we ask you address in a revised manuscript.  We are satisfied with most responses in your revised paper, but agree with the comments made by reviewer 1 and this this issue must be addressed.  

We therefore ask you to modify the manuscript according to recommendations of reviewer 1. 

[LINK]

Yours sincerely,

Lisa Stubbs

Associate Editor

PLOS Genetics

Hua Tang

Section Editor: Natural Variation

PLOS Genetics

Reviewer's Responses to Questions

**Comments to the Authors:**

Reviewer #1: I am satisfied with all responses to my reviews except the one related to FDR correction on GO analysis. I’m sorry, but false discovery rate correction is no less important in this instance. The authors of TopGO are probably right that the Benjamini-Hochberg method is inappropriate for the highly dependent p-values derived from GO analyses. Their assertion in their whitepaper that tests are already FDR-corrected by dint of the weighted method is dubious. If the TopGO authors attempted to shepherd this statement through peer review, I would venture to guess that they would receive substantial pushback from the statistical genomics community that solved this problem 20 years ago.

An FDR-correcting procedure known as Benjamini-Yekutieli (2001) was developed as an extension of Benjamini-Hochberg to conservatively estimate FDRs in the presence of dependencies between p-values. Applying this correction procedure is as simple as using the function p.adjust(pvalues, method = "BY") in base R. I think it reasonable to request that the authors apply this B-Y FDR correction procedure to the GO results. However, if they choose not to apply this procedure and retain the text as is, I would request that they report p-values for all GO categories in Supplemental Table S4. Presently, just the top 75 findings are reported. This does not allow the reader the opportunity to see the p-value distribution of the whole test set, which is what is necessary to truly assess the output.

Reviewer #2: The authors have done a great job with responding to the reviewers' comment and revising the manuscript. The paper is now well structured and clear, the results are interesting, and the interpretation is appropriate. Overall, this study is a nice contribution to the literature.

Reviewer #3: The authors claim single read RNAseq is acceptable and their lower read depth also sufficient but this is a well criticized method (Freedman et al. 2020)

Freedman, A.H., Gaspar, J.M. & Sackton, T.B. Short paired-end reads trump long single-end reads for expression analysis. BMC Bioinformatics 21, 149 (2020). https://doi.org/10.1186/s12859-020-3484-z

The lack of comparative analyses or comparative methodologies are of continued concern and it is disappointing that no attempt was made to related their findings the many other studies and gene lists available.

**Have all data underlying the figures and results presented in the manuscript been provided?**

Reviewer #1: Yes

Reviewer #2: Yes

Reviewer #3: Yes

PLOS authors have the option to publish the peer review history of their article (what does this mean?). If published, this will include your full peer review and any attached files.

Reviewer #1: **Yes: **Michael C. Saul

Reviewer #2: No

Reviewer #3: No

---

## [Editor Report · Decision Letter 2]

3 Aug 2021

Dear Dr Sheehan,

We are pleased to inform you that your manuscript entitled "Dynamic neurogenomic responses to social interactions and dominance outcomes in female paper wasps" has been editorially accepted for publication in PLOS Genetics. Congratulations!

Yours sincerely,

Lisa Stubbs

Associate Editor

PLOS Genetics

Hua Tang

Section Editor: Natural Variation

PLOS Genetics

Comments from the reviewers (if applicable):

**Data Deposition**

http://datadryad.org/submit?journalID=pgenetics&manu=PGENETICS-D-21-00338R2

**Press Queries**

---

## [Editor Report · Acceptance letter]

12 Aug 2021

PGENETICS-D-21-00338R2 

Dynamic neurogenomic responses to social interactions and dominance outcomes in female paper wasps 

Dear Dr Sheehan, 

We are pleased to inform you that your manuscript entitled "Dynamic neurogenomic responses to social interactions and dominance outcomes in female paper wasps" has been formally accepted for publication in PLOS Genetics! Your manuscript is now with our production department and you will be notified of the publication date in due course.

With kind regards,

Agnes Pap

PLOS Genetics

On behalf of:
